# Focus Your Attention (with Adaptive IIR Filters)

**Shahar Lutati** and **Itamar Zimerman** and **Lior Wolf**

The School of Computer Science, Tel Aviv University

shahar761@gmail.com

zimerman1@mail.tau.ac.il

wolf@cs.tau.ac.il

## Abstract

We present a new layer in which dynamic (i.e., input-dependent) Infinite Impulse Response (IIR) filters of order two are used to process the input sequence prior to applying conventional attention. The input is split into chunks, and the coefficients of these filters are determined based on previous chunks to maintain causality. Despite their relatively low order, the causal adaptive filters are shown to focus attention on the relevant sequence elements. The new layer is grounded in control theory, and is shown to generalize diagonal state-space layers. The layer performs on-par with state-of-the-art networks, with a fraction of their parameters and with time complexity that is sub-quadratic with input size. The obtained layer is favorable to layers such as Heyna, GPT2, and Mega, both with respect to the number of parameters and the obtained level of performance on multiple long-range sequence problems.

## 1 Introduction

Designing sequence models that capture short- and long-term dependencies is a central goal of sequence modeling. Besides performance, computational complexity also plays a part when dealing with long sequences. Although transformers (Vaswani et al., 2017) excel at tasks that involve short-range dependencies, their performance on data with long-range dependencies can be poor. For example, regardless of the high time complexity, on the Long Range Arena benchmark (LRA) (Tay et al., 2020) transformers perform poorly compared to other sequence models.

Another approach that has emerged to address long-range data processing is the utilization of regularized implicit global (long) convolutions. In this technique, convolutions are employed along the sequence dimension, enabling convolutions with a global receptive field. Initially, this approach was implemented through state-space layers (Gu et al., 2021b,a), which introduced a recurrent layer that could be efficiently computed via a global convolution. Recent research has explored alternative variations of global convolutions, including implicit (Poli et al., 2023) and regularized parameterization (Fu et al., 2023a; Li et al., 2022). These methods have demonstrated improved performance on tasks involving long-range dependencies and with sub-quantitative complexity. They have also shown effectiveness in enhancing long-range transformer capabilities (Ma et al., 2022; Zuo et al., 2022; Saon et al., 2023). However, it remains uncertain whether these models can scale up or function similarly to transformers across a diverse range of tasks (Vardasbi et al., 2023).

This work strives to efficiently integrate convolution-based sequence models and transformers, to provide a model that is capable of handling both short and long dependencies. The attempt to combine these components was first presented by Ma et al. (2022), who used a simple global convolution before each transformer block. This convolution is parameterized by the Exponential Moving Average (EMA) recurrent rule and can be seen as an IIR filter. In this work, instead of using first-order IIR filters, we introduce learnable adaptive IIR filters, which allow us to propose Focus, a layer that combines local attention and a novel type of regularized global convolution grounded on a hypernetwork that produces adaptive IIR filters.

**Our main contribution** is the focus layer, which has several unique properties: (i) We are the first to use data-dependent global filters, which are implemented by a global hyper-network mechanism that focuses local attention. (ii) In contrast to other works in the domain that employ FIR filters, it relies on IIR filters. (iii). We present an efficient and stable computation of those IIR filters. (iv) Theoretically, our layer is grounded in the theory of control systems, similar to state-space layers, which are built on the state-space model (SSM) of control theory. Furthermore, in Sec. 4 we show

that IIR filters are a generalization of SSMs and diagonal-linear RNNs, which have recently been recognized as remarkable long-range learning architectures (Gupta et al., 2022a; Gu et al., 2022; Orvieto et al., 2023; Gupta et al., 2022b; Saon et al., 2023; David et al., 2023). By drawing upon the extensive research conducted on IIR filters, our findings can provide additional insights into the effectiveness, stability, expressiveness, and initialization of those layers.

## 2 Background and related work

IIR filters, known as infinite impulse response filters, are digital filters that utilize feedback to generate an output signal. Their primary applications involve signal smoothing, filtering, and signal modification. These filters are extensively employed in various fields, such as audio processing, speech processing, and image processing. One notable advantage of IIR filters is their ability to achieve a significantly sharper roll-off in the transition region compared to an FIR filter of the same order. This is made possible by the presence of complex poles in the IIR filters, which enable them to attenuate frequencies more rapidly.

The state-space representation of an IIR filter is a convenient way to represent the filter's dynamics and to implement it in software. It consists of three parts: the state vector, the state transition matrix, and the output matrix. The state vector contains the filter's internal state variables, the state transition matrix describes how the state vector changes over time, and the output matrix describes how the output signal is computed from the state vector. Such representation are described in (Zhang et al., 2023)

**Learnable IIR Filters**   Since IIR filters are computationally efficient, yet expressive, it is natural to design IIR filters with Deep Learning. (Kuznetsov et al., 2020) proposes an approach to using traditional digital IIR filter structures inside deep-learning networks trained using backpropagation. The authors establish the link between such structures and recurrent neural networks and present three different differentiable IIR filter topologies. They compare the proposed topologies against each other and an established baseline and show that the proposed topologies can achieve better performance in some cases. Additionally, the authors present a simple Wiener-Hammerstein model, using differentiable IIRs as its filtering component,

and train it on a guitar signal.

**Global Convolutions**   The global convolution, also known as a long convolution, is a layer that applies scalar convolutions along the sequence dimension, enabling the handling of unrestricted 1-D sequences. Empirically, these layers have shown strong performance in tasks involving long-range dependencies, particularly in domains such as NLP (Dao et al., 2022b; Mehta et al., 2022; Wang et al., 2022), audio (Goel et al., 2022), speech (Saon et al., 2023), video (Islam et al., 2022; Wang et al., 2023), time-series analysis (Zhang et al., 2023) and more. Moreover, they exhibit computational efficiency as their cost is sub-quadratic. However, to achieve SOTA results, appropriate regularization is necessary. The approach of (Gu et al., 2021b,a; Ma et al., 2022; Li et al., 2022) incorporates a parameterization that inherently regularizes the kernel and decoupling sequence length from parameter count. (Romero et al., 2021; Poli et al., 2023) utilizes an implicit parameterization learned by FFNs operating on positional encodings, while (Fu et al., 2023a) explicitly regularizes the convolution kernels using squash or smooth operators.

**Long Range Transformers**   Transformers (Vaswani et al., 2017) have emerged as highly effective models for various tasks, but their widespread adoption has been limited by the quadratic cost of the self-attention mechanism and poor performance on long-range tasks. Researchers have pursued diverse approaches to overcome this challenge and to create efficient transformer architectures (Fournier et al., 2021; Tay et al., 2022). From the perspective of efficiency, techniques such as sparse attention (Child et al., 2019), low-rank attention (Wang et al., 2020; Winata et al., 2020), kernel-based attention (Choromanski et al., 2020), recurrent mechanisms (Hutchins et al., 2022; Dai et al., 2019), and efficient IO-awareness-based implementation (Dao et al., 2022a) proved efficient. From the perspective of effectiveness, (Yu et al., 2023; Ivgi et al., 2023) combines local and global attention models hierarchically, enhancing the model's ability to handle extensive context Other techniques employ global memory-based Attention (Gupta and Berant, 2020; Al Adel, 2022; Al Adel and Burtsev, 2021), and (Zhou et al., 2022) applies attention in the frequency domain to expand long-range capabilities.

**Hyper Networks**   A hypernetwork (Ha et al., 2016) is a function that maps a set of inputs to

a set of weights, which are used as the parameters of a "primary network". Hypernetworks have been shown to be effective for a variety of tasks, including, for example, image classification (Lutati and Wolf, 2023), natural language processing (He et al., 2022), and speech recognition (Szatkowski et al., 2022). They have also been shown to be able to improve the performance of neural networks on meta-learning tasks, such as few-shot learning (Bertinetto et al., 2016), continual learning (Von Oswald et al., 2019), and neural architecture search (Zhang et al., 2019).

**Adaptive Filtering**   Adaptive filtering is a technique used to improve the quality of a signal by removing noise or interference. Adaptive filters are able to adapt to changes in the signal or the environment, making them well-suited for a variety of applications.

One common technique used in adaptive filtering is the short-time Fourier transform (STFT), which provides a time-frequency representation of a signal. It enables the analysis of time-varying properties of a signal by dividing it into short-time windows and applying the Fourier transform to each window. The STFT reveals the distribution of frequency content over time, which allows the adaptive filter to track the frequency content of the signal and adapt its coefficients accordingly. However, STFT introduce non-causal implementation due to overlapping time-bins. To mitigate it, we introduce chunked-FFT, a degenerate form of the STFT.

Recent research in AI has focused on using deep learning to improve the performance of adaptive filters. For example, deep learning has been used to improve the performance of adaptive filters for noise cancellation (Zhang and Wang, 2021), echo cancellation (Haubner and Kellermann, 2022), and equalization (Zhou et al., 2020). Deep learning has also been used to develop new adaptive filter architectures that are more robust to noise and interference (Alwan and Hussain, 2022). Revach et al. (2022) demonstrate how deep learning can be used to improve the performance of Kalman filtering (Kalman, 1960), a classical control algorithm.

## 3   Method

### 3.1   Overview

We start by discussing the main design choices of our architecture.

**Chunking and the combination of local and global models**   Given the quadratic complexity of transformers, chunking is a common practice for computing short-range attention efficiently. However, despite excelling in short-range tasks, full-length transformers often struggle to handle long-range dependencies and often perform comparably to local-attention-based transformers (Xiong et al., 2021). Recent studies have demonstrated that a combination of local and global transformers can achieve state-of-the-art performance on long-range tasks (Ivgi et al., 2023; Yu et al., 2023; Hutchins et al., 2022). Inspired by these findings, we introduce local attention as the local model, which is combined with a novel type of global convolution as the global model. Furthermore, in contrast to (Hutchins et al., 2022; Bulatov et al., 2023), our global model does not use recurrent computations, since it severely restricts parallelization.

**Adaptive IIR Filters**   In MEGA (Ma et al., 2022), it was demonstrated that incorporating an EMA at the beginning of each transformer block improves transformer performance in long-range tasks. EMA can be viewed as a convolution operation using simple first-order IIR filters. Motivated by this finding, we adopt a more versatile and expressive convolution approach that utilizes adaptive filters generated by a hypernetwork. Since the hypernetwork is an integral part of our global model, it employs global convolutions. Specifically, the regularized global convolution of (Fu et al., 2023a) is used, as the most straightforward option. A common challenge with hypernetworks is ensuring relatively small output sizes. In this regard, leveraging IIR filters, which have only a few parameters, is a reasonable choice.

### 3.2   The Focus Layer

In this section, we describe the focus attention head, our primary contribution. This head is integrated into the MEGA backbone, as visualized in Fig 1. Let $x$ be the input for the focus layer, where $x \in \mathcal{R}^{L \times D}$, $L$ is the sequence length, and $D$ is the input's dimension. Our method, termed Focus, utilizes the foundations of adaptive filtering theory to cope with very long stochastic sequences. Given the seasonality of the sequence, the resolution of the FFT is determined by its size in each time-bin. Denote the size of the FFT in a single time-bin as $NFFT$.

The first component of the Focus layer is the hypernetwork, $H$. The output of $H$ is $\Theta$, which

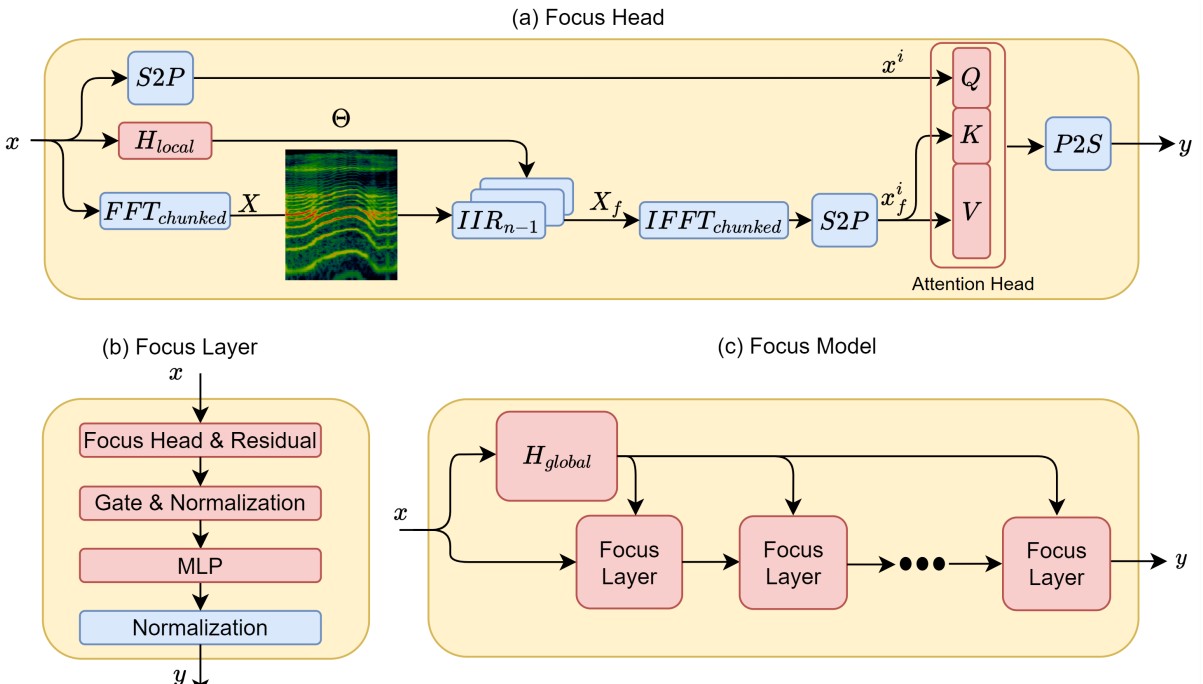

(a) Focus Head

(b) Focus Layer

(c) Focus Model

Figure 1: Focus Architecture: (a) The architecture of a single head. (b) The obtained layer. (c) The entire model. The architecture of the model and layer are defined similarly to MEGA (Ma et al., 2022). Blocks in blue are not learned, while blocks in red are learned parameters. $S2P$ (serial to parallel) and $P2S$ (parallel to serial) are the chunking and the de-chunking operations, respectively.

is the set of IIR kernels used for the forward processing of the sequence. $\Theta$ has a dimension of $Nbins \times D \times F \times 2$, denoting $F$ kernels, each with a kernel size of two for $D$ feature channels. The kernel is unique per time-bin, $Nbins$, which makes the filter adaptive to changes over time.

$$\Theta = H(x). \quad (1)$$

$H$ has two main components. The first is a shallow global convolution (Fu et al., 2023b) based submodel that is followed by adaptive max pooling (over each feature channel) (Pytorch, 2023) with a size of $O \times Nbins$, where $O$ is the oversampling factor.

$$e = \text{MaxPool}(\text{GlobalConv}(x)) \quad (2)$$

where $e$ is the embedding from processing $x$ using the global convolution layer. This computation can be shared across multiple Focus layers and can be split into local ($H_{\text{local}}$) and global ($H_{\text{global}}$) subcomponents, as in most of our experiments, thus reducing substantially the computational cost. Furthermore, the embedding is permuted such that the feature space has the size $O$ while $Nbins$ is added to the batch dimension for parallel computing.

The second component of $H$ is a 2-layer MLP with sigmoid activations that maps the embedding $e$ to a tensor with size $Nbins \times D \times F \times 2$. With mapping of latent dimension $O$ to $2 \cdot F$

$$\Theta = MLP(e), \quad (3)$$

where $\Theta$ is the IIR kernel, with size $Nbins \times D \times F \times 2$. $MLP$ is the forward MLP mapping, as described above. Since $H$ is a hypernetwork, the initialization of the last layer of the MLPs follows (Chang et al., 2020). The rest of the layers follow the Xavier initialization (Glorot and Bengio, 2010). The input $x$ is split into non-overlapping time bins, where each time bin is passed through the FFT of the size $NFFT$. Denote the input in the r-th time bin as $x_r$.

$$X[\omega, r] = FFT(x_r), \quad (4)$$

where $\omega$ is the normalized frequency variable, sampled evenly on $2\pi$ range, and $r$ is the index over the different time bins.

**A note about causality**    To maintain a fully causal model that is applicable to auto-regressive tasks, each time bin is processed on its own and is not overlapped with other time bins. In addition, $\Theta$ is shifted right by one time bin, such that the

sequence at time bin $i$ is processed by the kernel computed from time bin $i - 1$.

For each time-bin index, the corresponding IIR filter is applied. The IIR filter of order 2 has the following frequency response, denote it as $IIR_{imp}$

$$IIR_{imp}(f) = \frac{1}{1 + \Theta[0] \cdot e^{-j \cdot 2\pi f} + \Theta[1] \cdot e^{-j \cdot 4\pi f}}, \quad (5)$$

Since a Sigmoid activation is used for the last layer of $H$, it is guaranteed that $\Theta$'s elements are positive real numbers smaller than 1. Further analysis and reasoning behind the specific selections made are presented in Sec. 4.

Recall that in the frequency domain, the equivalent of filtering is multiplying with the conjugated impulse response,

$$X_f = X \circ IIR_{imp}^*, \quad (6)$$

where the conjugation is denoted by a star and $\circ$ is the elementwise (Hadamard) multiplication. The hyper-dimension $F$ defined earlier as the filter-bank size is collapsed via regular sum operation, denote the collapsed tensor as $X_c$

$$X_c = \mathbf{1} \cdot X_f, \quad (7)$$

where $\mathbf{1}$ is an all-ones vector with size $1 \times F$. The collapsed tensor is the short-time Fourier representation of the original sequence filtered with adaptive filter kernels. The frequency representation goes through the inverse chunked Fourier transform (IFFT), to obtain the time-domain sequence.

$$x_f = IFFT(X_c) \quad (8)$$

The time-domain sequence has the same dimensions as the original sequence, yet, by applying an adaptive filter to it, we furnish it with an induction bias that helps smaller context attention head to cope with complicated tasks.

The sequence is split into $C$ separate non-overlapping chunks. Denote the chunk length as $M$, such that $L = MC$. We denote the chunk of signal with uppercase $i$, and use square brackets for indexing, starting with index 0, as follows

$$x^i = (x[iM], x[iM+1], \dots, x[(i+1)M-1]) \quad (9)$$

All chunks are processed in parallel with the same small attention head,

$$y^i = Atten(Qx^i, Kx_f^i, Vx_f^i), \quad (10)$$

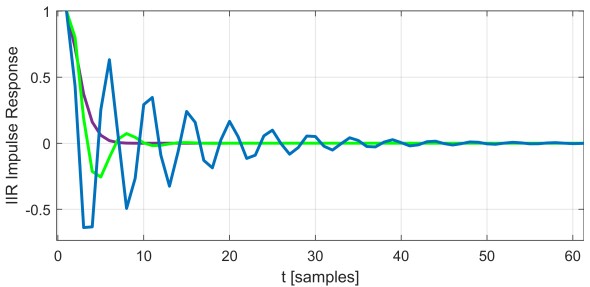

Figure 2: Filter responses for three random filters with the specific denominator structure of Eq. 16.

where $Q$, $K$, $V$ are the query, key, and value matrices that map each chunk to their latent corresponding space. Note that the attention head is causal, as it uses lower triangle masking. The chunks, $y^i$, are rearranged to form a complete signal, with sequence length $L$, denote it as $y$. Following (Ma et al., 2022), the output of the Focus layer, $y$, passes through reset gate $\gamma$, the update gate $\psi$. Specifically,

$$\gamma = SiLU(x_f W_\gamma + b_\gamma) \quad (11)$$

$$\phi = sigmoid(x_f W_\phi + b_\phi) \quad (12)$$

$$z = SiLU(x_f W_h + (\gamma \circ y)U_h + b_h), \quad (13)$$

where $W_\gamma$, $W_\phi$ and $W_h$ are learned matrices with size $D \times D$. $b_\gamma$, $b_\phi$, and $b_h$ are learned biases with size $D$. SiLU stands for the sigmoid linear unit (Elfwing et al., 2018).

The final output is the gated summation of the gated attention and the input sequence,

$$o = \phi \circ z + (1 - \phi) \circ x \quad (14)$$

## 4 Analysis

**IIR and FIR Filters**  IIR (Infinite Impulse Response) and FIR (Finite Impulse Response) filters are two commonly used types of digital filters with distinct characteristics. The main difference between them lies in their impulse response and filtering properties. FIR filters have a finite impulse response, meaning that the filter output is based solely on a finite number of past input samples. In contrast, IIR filters have an infinite impulse response, allowing the filter output to depend on both past and future input samples.

One advantage of IIR filters is their ability to achieve a desired frequency response with fewer coefficients compared to FIR filters. This makes IIR filters more computationally efficient, requiring fewer calculations and lower memory requirements.

Consequently, in control feedback systems where real-time operation and computational efficiency are crucial, IIR filters are often preferred.

Additionally, IIR filters can exhibit higher selectivity and sharper roll-off in the frequency domain compared to FIR filters. This characteristic can be advantageous in control feedback systems, where precise control over specific frequency components is necessary.

**Stability** IIR filters can be more sensitive to quantization errors and can be prone to instability if not properly designed. The presence of feedback loops in control systems can further impact stability considerations. Therefore, careful attention must be given to stability analysis and appropriate filter design techniques to ensure reliable performance.

The filter can be described in the frequency domain as a rational polynomial function of the complex exponent $e^{-j2\pi f}$. Denote the complex exponent as $S$.

$$S = e^{-j2\pi f} \qquad (15)$$

A second-order system can be described as follows,

$$IIR(S) = \frac{1}{aS^2 + bS + 1} \qquad (16)$$

Specifically solving for general $b, a$ gives,

$$IIR(t) = \alpha e^{t\left(-\frac{b}{2a} - \frac{\sqrt{b^2 - 4a}}{2a}\right)} + \beta e^{t\left(-\frac{b}{2a} + \frac{\sqrt{b^2 - 4a}}{2a}\right)}, \qquad (17)$$

where $\alpha$ and $\beta$ are normalizing factors. Denote the term under the square root as discriminant, $\Delta$. For any $b \geq 0$ and $a \geq 0$ the following holds:

$$\Delta \leq b^2 \quad \forall \{a, b\} \geq 0. \qquad (18)$$

In order to guarantee stability, both exponents should decay, which leads to the requirement that the real part must be negative.

$$Re\{-\frac{b}{2a} \pm \frac{\sqrt{\Delta}}{2a})\} \leq 0. \qquad (19)$$

This can be achieved if $b$ is positive. In the scenario where $\Delta \leq 0$, the exponents have an imaginary part, causing it to oscillate. This is called an under-damped response. This response is stable, yet more expressive than Exponential Moving Average (EMA) filters, as found in MEGA (Ma et al., 2022). The frequency of the sine, $\omega$, in this case is

$$\omega = \frac{\sqrt{\Delta}}{2a}, \qquad (20)$$

and the time domain impulse response reads,

$$IIR(t) = \gamma e^{-t\frac{b}{2a}} sin(\omega t + \phi), \qquad (21)$$

where $\gamma$ is the normalizing factor, and $\phi$ is the phase from aggregating both sine and cosine functions with the same frequency. Note that for orders above 2, there is no simple condition that guarantees that the real part will be negative and the response stable.

To demonstrate the oscillating behavior of the generated IIR filters, the time-domain impulse responses of some random kernels are drawn in Fig. 2. In this plot, the purple kernel acts as EMA, while other kernels have a more complicated response. However, all filters decay as time increases, which leads to a stable response and has been identified by Li et al. (2022) as an essential property for capturing long-range dependencies.

**Time Complexity** The global conv time complexity is

$$\text{GlobalConv} \approx O(Llog(L) \cdot D) \qquad (22)$$

This is due to the FFT and IFFT that the signal is passed through. This computation is done only once and is shared through multiple layers of Focus. The MLP is mapping between $D$ and $F$.

$$MLP \approx O(DF) \qquad (23)$$

The chunked-FFT complexity depends on the size of the FFT used ($NFFT$). Denote the size of a single time bin as $R$,

$$R = \frac{L}{Nbins} \qquad (24)$$

The time complexity of a single time bin is $O(Rlog(R))$. The total time complexity of the chunked FFT reads,

$$\text{FFT}_{\text{chunked}} \approx O(Llog(R)) \qquad (25)$$

Next, the time complexity of the attention head depends on the size of the context length $M$,

$$\text{Atten} \approx O(CM^2D + CMD^2) \qquad (26)$$

The total time complexity of the Focus layer is, therefore,

$$\text{Focus} \approx O(Llog(L) \cdot D + CM^2D) \qquad (27)$$

where we neglected smaller terms when the sequence length is large (greater than dimensions).

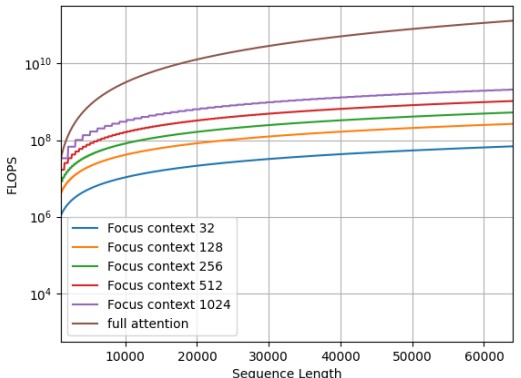

Figure 3: Time Complexity of the Focus layer and of Attention, increasing sequence length

Recalling that $L = MC$, and rearranging terms, we have,

$$\text{Focus} \approx O(DL \cdot (log(L) + M)) \qquad (28)$$

obtaining sub-quadratic time complexity with respect to input sequence length. A visual comparison of overall complexity versus the standard attention head is depicted in Fig. 3.

**Expressiveness** An emerging class of diagonal linear RNNs (Orvieto et al., 2023; Gupta et al., 2022b) recently achieved near SOTA results in several long-range tasks. They include complex and real variants, as well as diagonal state-space layers (Gupta et al., 2022a; Gu et al., 2022). The following recurrent rule describes each channel of those layers:

$$s[t] = As[t-1] + Bx[t], \quad y[t] = Cs[t] + Dx[t] \qquad (29)$$

where $s[t]$ is the recurrent state at time $t$. By isolating $s[t-1]$, we can rewrite Eq. 29 as follows:

$$s[t-1] = \frac{1}{C}y[t-1] - \frac{D}{C}x[t-1] \qquad (30)$$

$$y[t] = CAs[t-1] + (CB+D)x[t] = \qquad (31)$$

$$Ay[t-1] + (CB+D)x[t] + ADx[t-1]$$

Recall that the differential equation of an IIR filter of order 2 can be represented as follows:

$$y[t] = b_0 x[t] + b_1 x[t-1] + b_2 x[t-2] - \qquad (32)$$

$$a_1 y[t-1] - a_2 y[t-2]$$

By substituting the values of $b_0 = CB+D$, $b_1 = AD$, $a_1 = -A$, $b_2 = a_2 = 0$, it becomes evident

that the IIR filter can be constrained to a linear SSM. In machine learning, $D$ is often omitted in SSMs or diagonal RNNs, since it can be seen as a parameter-based skip-connection. In this case, the SSM can be represented by an IIR filter of order 1.

As mentioned earlier, higher-order filters can introduce stability issues. Therefore, our decision to utilize IIR filters of order 2 is justified, as we opt for the most expressive IIR filters that still maintain stability during training.

## 5 Experiments

Below we present experimental results for the proposed Focus layer. In addition to our full method, we introduce an ablation to evaluate the importance of adaptive filtering, in which instead of the hyper-network $H$, the IIR filters are conventional learned parameters. This ablation is denoted by "Focus-H".

### 5.1 In-context learning

In order to evaluate our method relative to other state-of-the-art long-range architectures, such as (Poli et al., 2023), (Dai et al., 2019), the associative recall synthetic task is evaluated. The associative recall task was first introduced in (Elhage et al., 2021) and is part of a number of simple yet informative tasks that test the capabilities of the model in processing long-range sequences.

In the associative recall task, each string is formed by concatenating key-value pairs sampled randomly from a dictionary. The model should output the correct value given a singular key, regardless of whether the key is in the long sequence.

Similarly to Poli et al. (2023), we employ the associative recall task in order to explore the memory capabilities of our model.

In all synthetic data experiments the same shared hyperparameters are used, with the exception of the sequence length. The hyperparameters are depicted in in Appendix A. The AdamW optimizer (Loshchilov and Hutter, 2018) is used.

As can be seen in Tab. 1, our model is able to obtain an accuracy of 100% for all sequence lengths, without overfitting, despite the low number of examples (2000), and with no memory explosion thanks to linear scaling with input size. These results show that the Focus mechanism is able to improve the performance of regular transformers to be on par with Heyna (Poli et al., 2023), with smaller footprint. In addition, the ablation experiment shows the importance of adaptive filtering,

Table 1: Test accuracy (%) for associative recall on long sequences of length $L$ and a vocabulary size of 30. NF - not feasible to test. NR = not reported.

| L | Focus | Focus-H | Hyena | FlashTransformer | Transformer |
|---|-------|---------|-------|------------------|-------------|
| 30 | 100.0 | 100.0 | 100.0 | 100.0 | 100.0 |
| 1K | 100.0 | 98.0 | 100.0 | 95.0 | 100.0 |
| 8K | 100.0 | 85.3 | 100.0 | NR | NF |
| 32K | 100.0 | 34.6 | 100.0 | 32.4 | NF |
| 64K | 100.0 | 28.0 | 100.0 | 26.7 | NF |

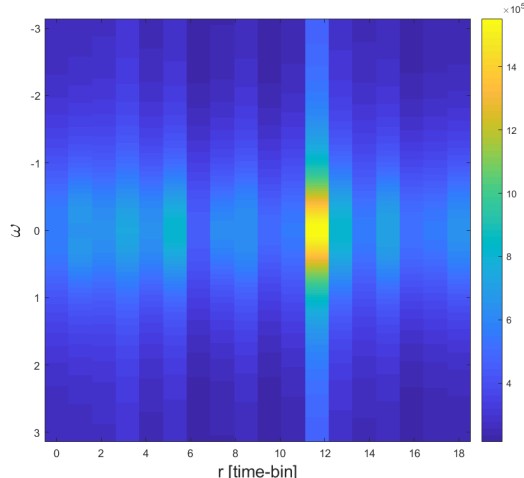

Figure 4: Frequency Response of IIR kernels, for 1K sequence split into 18 time bins. The important key is found in the 12th time bin.

i.e. estimating the filter kernels online to focus the attention mechanism on important sub-sequences where the results degrade in the ablation due to the greater sequence length.

The associative recall task is used not only for benchmarking, but also to gain insights into the adaptive filtering mechanism. As can be seen in Fig. 4, the frequency response of the adaptive filtering is plotted for this task. In this specific run, the important key is found in the 12th time bin. The frequency response of the IIR filters is almost 5 orders of magnitude higher than for nearby time bins. This effect demonstrates the "Focus" mechanism. Note that using only 2 parameters for the IIR kernel, the filters are able to differentiate between important and unimportant time bins with 5 orders of magnitude. This supports our design choice of IIR filter, seeing that with a kernel size as low as 2 the filter is still sharp enough.

## 5.2 Language Modeling

The enwiki8 dataset is a byte-level dataset consisting of the first 100 million bytes of a Wikipedia XML dump. It is a commonly used dataset for benchmarking character-level language models.

The Text8 dataset is a corpus of text used for training and evaluating language models. It is a subset of the Wikipedia dump from March 2006 and consists of 90 million characters. The text is tokenized and lowercased, and each token is assigned a unique id.

The metric used to evaluate language models on enwiki8 and Text8 is bits per character (BPC). The lower the BPC, the better the language model. To compute the BPC the average cross-entropy is computed in the log2 basis

$$BPC = \frac{1}{L} Log_2 \text{CrossEntropy}(P, \hat{P}), \quad (33)$$

where $P$ is the target distribution, and $\hat{P}$ is the output distribution. $L$ is the sequence length. To maintain the same capabilities such as Mega (Ma et al., 2022), we used 8 layers of the Focus layer, with a hidden dimension of 1024 and an input dimension of 512.

The enwiki8 results are reported in Tab. 2. Evidently, Focus outperforms both Mega (Ma et al., 2022) and Transformer XL (Dai et al., 2019), despite a much lower number of parameters, performing on-par with GPT2(Radford et al., 2019) (zero-shot) but with a fraction of its parameters and substantially less FLOPS. The same occurs for the Text8 dataset, reported in Tab. 3. While the ablation is inferior to Transformer-XL, with the full method Focus is on par with GPT2.

## 5.3 1-D image classification

We evaluated our model on the sequential MNIST task, a challenging problem that requires models

Table 2: BPC for enwiki8 dataset.

| Model | #params | BPC |
|---|---|---|
| Transformer XL (Dai et al., 2019) | 277M | 0.99 |
| Mega (Ma et al., 2022) | 39M | 1.02 |
| GPT2 (Radford et al., 2019) | 1542M | 0.94 |
| Focus-H (ablation) | 21M | 1.06 |
| **Focus** | 22M | **0.94** |

Table 3: BPC for text8 dataset.

| Model | #params | BPC |
|---|---|---|
| Transformer XL (Dai et al., 2019) | 277M | 1.08 |
| GPT2 (Radford et al., 2019) | 1542M | 0.98 |
| Focus-H (ablation) | 21M | 1.10 |
| **Focus** | 22M | **0.98** |

Table 4: Accuracy for sMNIST, pMNIST

| Model | sMNIST | pMNIST |
|---|---|---|
| Transformer | 98.9 | 97.9 |
| S4 (Gu et al., 2021a) | 99.6 | 98.7 |
| Focus-H (ablation) | 98.9 | 98.0 |
| **Focus** | **99.7** | **98.8** |

| Model | Inference time | Memory |
|---|---|---|
| Transformer | x1 | x1 |
| S4 | x1.58 | x0.43 |
| MEGA | x1.49 | x0.57 |
| **Focus** | x1.75 | x0.38 |

Table 5: Comparison of inference speed and peak memory consumption for related models.

## 6 Conclusions

Attention models are extremely powerful for modeling sequences, as demonstrated by the seminal work of Vaswani et al. (2017). Indeed, transformers have revolutionized the way deep learning is practiced, leading to unprecedented performance across almost all studied AI domains.

However, transformers have quadratic complexity in the sequence length, which can impact their efficiency, and often struggle to perform optimally in tasks that involve long-range dependencies (Tay et al., 2020). In this work, we present a dynamic filtering approach that enables us to subsequently employ attention within much shallower architectures. As our ablation shows, the dynamic nature of these filters is crucial to the success of the layer. Similarly crucial is the use of IIR filters, and we analyze the regime in which these are stable.

## 7 Limitations

While this paper presents promising results, there are a few limitations to consider. Firstly, although we are the first to utilize IIR filters for long-range tasks, we have not examined sequence models solely based on IIR filters. Additionally, we have not investigated the impact of different types of hyper-global convolution on performance. Furthermore, IIR filters can be computed using recurrent rules or convolution. While the convolutional view is more natural for training, the recurrent view presented in Eq. 32, can be leveraged for efficient auto-regressive generation. This can lead to a significant reduction in the time and space complexity of the layer during inference, which is beneficial for real-time applications.

## 8 Acknowledgments

This work was supported by a grant from the Tel Aviv University Center for AI and Data Science (TAD). It is part of a PhD research conducted by the first author.

to capture long-range dependencies. Permuted MNIST is a variant of MNIST where the order of pixels in each image is scrambled, making the task more challenging. Following S4, we use a hidden dimension of 512 but to save resources, we use 6 layers and not more. The results are listed in Tab. 4. Focus has an accuracy of 99.7% (98.8%) on unpermuted (permuted) MNIST, outperforming the transformer and S4.

### 5.4 Efficiency Comparison

To assess the efficiency of the Focus layer, we measure peak memory usage and inference speed. We compare several related models on the association recall task. This task involves processing a sequence of 1K tokens, which represents the maximal fit in memory for transformers. The results are presented in Tab. 5. As can be seen, Focus exhibits the lowest peak memory consumption, using only 38% of the memory consumed by the Transformer model. However, its inference speed is slightly slower than the other methods.

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

# A  Hyperparameters

The hyperparameters for the associative recall task are provided in Tab. 6. Hyperparameters that differ in other experiments are reported in the respective sections. For example, in Sec. 5.2, eight layers of the Focus layer are used, with a hidden dimension of 1024 and an input dimension of 512, in order to compare with Mega on similar terms.

Table 6: Hyperparameter settings for the synthetic associative recall task

| | |
|---|---|
| Optimizer | AdamW |
| Optimizer momentum | $\beta_1, \beta_2 = 0.9, 0.98$ |
| Vocabulary Size | 30 |
| NFFT | $L/4$ |
| F | 1 |
| C | 32 |
| Learning Rate | 1E-4 |
| Batch Size | 32 |
| Num Samples | 2000 |
| Warmup epochs | 10 |
| Number of Layers | 2 |
| Width | 64 |