# OpenReview forum: "Focus Your Attention (with Adaptive IIR Filters)"
_EMNLP/2023/Conference — EMNLP 2023 Main_

### Official Review · Reviewer_EHXv · 2023-07-29

**Soundness:** 3

**Excitement:**

4: Strong: This paper deepens the understanding of some phenomenon or lowers the barriers to an existing research direction.

**Paper Topic And Main Contributions:**

The paper presents a new attention layer in which dynamic Infinite Impulse Response (IIR) filters of order two are used to process the input sequence prior to applying conventional attention. It lead to a strong reduction of parameter and sub-quatric computational complexity through chunking of the sequence. The aims of the author to combine system and signal processing insights to attention layers. It provide stability proof, low computational complexity and experimental evaluation on several datasets.

**Questions For The Authors:**

(A) What are the impact of the new parameters brought by your approach e.g. Nbins, F, H,...?

(B) Why are you applying signal processing technique only to the key and the value, and not to the query? How does it impact the value  output of the attention layer?

(C) How does your approach compare in terms computational complexity to MegaBytes, and sparse transfromer used in GPT-3?

Yu, Lili, et al. "Megabyte: Predicting million-byte sequences with multiscale transformers." arXiv preprint arXiv:2305.07185 (2023).

Rewon Child, Scott Gray, Alec Radford, and Ilya Sutskever. Generating long sequences with sparse transformers, 2019.


**Reasons To Accept:**

The strength of the paper are:

* The idea is novel and bridge the gap between system/signal processing to transformer. While text-based task will benefit from such layers. I believe this is likely to be more influencial for audio-based (ASR, TTS,...) task.
* Applies approach in two domains: (text, images_ accross 4 tasks.
* Computation complexity of the approach is given.
* A stability proof is provided, but unclear overall how it impact the algorithm in practice.



**Reasons To Reject:**

The weakness of the paper are:

* The method section is expecting advanced knowledge on various system and signal processing. The link between the signal processing and the impact within the attention network is not necessarily obvious. There is a lack of interpretations of why some design choices have been made.
* Lack of statistical analysis of the results. There is not even a standard deviation of the metrics.
* Unclear impact of the parameter brought by this new approach.
* Quite a few of the baseline are old: most are from 2019, one from 2021 and one from 2022. For instance, it uses GPT-2 as a baseline, where GPT3, GPT3.5 or equivalent would be more suitable.
* We are missing ablations studes on the components added by the IIR filter, e.g. applied on the key or only the value or added parameters, other type of filters,...



**Reproducibility:**

3: Could reproduce the results with some difficulty. The settings of parameters are underspecified or subjectively determined; the training/evaluation data are not widely available.

**Reviewer Confidence:**

3: Pretty sure, but there's a chance I missed something. Although I have a good feel for this area in general, I did not carefully check the paper's details, e.g., the math, experimental design, or novelty.

**Typos Grammar Style And Presentation Improvements:**

I would like to have more interpretation of the presented method, and if possible ablation study to highlight the benefit of the approach.

---

> ### Author Rebuttal · Authors · 2023-08-29
>
> Thank you for the comprehensive review and the constructive feedback.
> Addressing the issues raised as weaknesses one by one:
> 1. Advanced knowledge: In section 3 (“Methods”; L334) we refer to section 4 (“Analysis”) for
> further explanations, intuitions, and proofs, since, as the reviewer states, there is an advanced
> usage of knowledge from signal processing and control theory. We want the paper to be
> approachable to all members of the AI community, and thus we wrote section 4 in a way
> that delves into causality, stability, and IIR general formulation. In the final version of the
> paper (where an extra page is available), we will add text about intuition, to make the paper
> even more readable.
> 2. Statistical analysis: While previous work did not provide data beyond mean performance
> numbers, we will add the standard deviation to the experiments. Below is a sample table of
> PMNIST benchmark with five different seeds. As written in lines L623-L625, we were able
> to be on par with a Transformer with half the number of layers and half the number of
> parameters.
> | Model | sMNIST | pMNIST |
> |-----------------------|------------|------------|
> | Transformer | 98.90 | 97.90 |
> | S4 (Gu et al., 2021a) | 99.60 | 98.70 |
> | Focus-H (ablation) | 98.90±0.05 | 98.00±0.11 |
> | Focus | 99.70±0.08 | 98.80±0.12 |
> 3. Hyperparameters: The number of hyperparameters is similar to Mega, for example. The
> meaning of each hyperparameter is rather intuitive. When $Nbins\rightarrow1$ our method
> degenerates into the Mega method, when $Nbins\rightarrow T$ (where T is the sequence length) the
> method degenerates to a Transformer (since no temporal filtering is applied). The trade-off
> between being adaptive to fast changes in the sequence and maintaining context along very
> long sequences is controlled by this parameter.
> The “F” parameter controls the dimensionality of the network, with sufficient expansion that
> leads to generalization.
> The hypernetwork H adds adaptivity to the network, which, as we show, is important in long
> sequences since single-static filters as in Mega fail to produce meaningful features with
> different time scale events. This is shown in the ablation “-H” in tables 2-4.
> 4. Newer GPT baselines: The results of all baselines in Tab. 2 are taken from Mega [Xuezhe
> et al., 2023]. The training procedure for Focus (our layer) is exactly the same as for Mega
> and Transformer XL [Dai et al., 2019]. GPT2 is trained with _extra data_, and performs
> zero-shot, as detailed in [Radford et al., 2019], see L611-L612. It is given to put the number
> is context, not as a direct comparison.
> While there are a lot of challenges associated with scaling up models to GPT3, GPT3.5
> sizes, this is not our focus. In our work, we show, for the first time, how signal processing
> tools and theorems, combined with recent advances in long-range modeling (tackling long
> sequences with state-space models) can yield slim and effective models that surpass more
> than 2 orders of magnitude larger models. Thus the comparison to bigger, higher-capacity
> networks is out of the scope of this work.
> Needless to say, we compare to the state-of-the-art layers in the field of long-range
> dependencies such as Mega and Hyena.
> 5. More variants: In order to maintain the time resolution of the input sequence, at least one of
> the variables of the transformer (key or value) needs to be unchanged, otherwise, the result
> will be distorted. In section 4, we show that the most general filter to represent state-space
> modeling, with simple guarantees for causality and stability, is IIR of order 2. Other
> formulations, if not equivalent, simply do not converge (exploding gradients) or lead to
> non-causal reactions. In the full 9-page version we will add a few examples of those failed
> alternatives and show why it is important to have this specific formulation and order of
> filters.
>
> To the three questions asked by the reviewer:
>
> A. The signal is chunked to Nbins and this is the minimal window that the focus adaptive
>      filtering addresses. For $Nbins\rightarrow1$ the method degenerates into Mega formulation. For
>      $Nbins\rightarrow T$  the filter kernel size is one, and the method degenerates to a vanilla transformer. By
>      using the division of Nbins one controls the degree of adaptivity to non-stationary signals of
>      the algorithm. F is the number of channels, similar to the number of channels in CNN.
>
> B. The reasoning behind this choice (see also point 5 above) is to conserve at least one input
>      of the transformer to be with the same time resolution as the original signal, without
>      changing its spectrum. By using a multiresolution approach, our method effectively utilize the full bandwidth of the original data.
>
> C. In terms of complexity, our method is more efficient than these methods. MegaBytes is the
>      most efficient out of those mentioned by the reviewer and still has twice the complexity than
>      focus (MegaBytes was published on arXiv right before the EMNLP deadline).
>      The underlying cause for our efficiency is that for each example, we compute its
>      corresponding FFT representation only once. We note that this computational burden would
>      become negligible for deeper architectures.

---

### Official Review · Reviewer_Wq8Q · 2023-08-01

**Soundness:** 3

**Excitement:**

3: Ambivalent: It has merits (e.g., it reports state-of-the-art results, the idea is nice), but there are key weaknesses (e.g., it describes incremental work), and it can significantly benefit from another round of revision. However, I won't object to accepting it if my co-reviewers champion it.

**Paper Topic And Main Contributions:**

This paper introduces a novel layer that incorporates dynamic Infinite Impulse Response (IIR) filters of order two to process input sequences before applying conventional attention. The filters are input-dependent and causal, preserving causality by determining their coefficients based on previous chunks. Despite their low order, these adaptive filters effectively focus attention on relevant sequence elements. The layer performs comparably to state-of-the-art networks while using significantly fewer parameters and boasting a time complexity that is sub-quadratic with input size.

**Reasons To Accept:**

1. The introduction of the background and related work is quite comprehensive.
2. The analysis of method stability, time complexity, and expressiveness is detailed and comprehensive.

**Reasons To Reject:**

1. The novelty of the method is not easy to determine, as it appears to be a combination of existing approaches.
2. The excessive number of equations in the paper makes it challenging to comprehend, and it occupies significant space, reducing the presentation of experiments and ablation studies. Consider minimizing the use of equations to improve readability and allocate more space for experimental results and ablation studies.
3. The experimental section lacks sufficient validation, as it utilizes relative simple tasks and datasets, and a limited number of baseline models, which results in unreliable and insufficiently convincing results. Additionally, the evaluation metrics are too simplistic, focusing solely on performance and parameter count. It is recommended to refer to Mega's experimental section[1], where they employ five different data types for sequence modeling tasks, and incorporate diverse baseline models and evaluation metrics (including training speed and peak memory consumption comparisons, in addition to performance assessment).
4. The citation format is not standardized (please try to cite formal published versions and avoid citing Arxiv, such as Mega [1], and others, please check accordingly).

[1] MEGA: MOVING AVERAGE EQUIPPED GATED ATTENTION   ICLR2023

**Reproducibility:**

4: Could mostly reproduce the results, but there may be some variation because of sample variance or minor variations in their interpretation of the protocol or method.

**Reviewer Confidence:**

3: Pretty sure, but there's a chance I missed something. Although I have a good feel for this area in general, I did not carefully check the paper's details, e.g., the math, experimental design, or novelty.

---

> ### Author Rebuttal · Authors · 2023-08-29
>
> Thank you for the comprehensive review and the constructive feedback.
> 1. Our method builds upon the key insights of recent state-of-the-art methods,
>     each of which is a seminal work in its own right but it is also the first method to
>     offer a control-theory theorem showing that the best stable and causal
>     state-space filter for the long-range setting is an IIR filter of order 2. Such filters
>     were not considered before in the construction of attention or other layers.
>     While we do not claim to have invented exponential moving average or
>     transformers, the insights and mathematical framework presented in our paper
>     shed new light on these methods and demonstrate, for the first time, how
>     state-space models can be incorporated in a stable and causal way with the
>     expressive power of transformers.
> 2. We would follow your advice, and try to reduce some of the space allocated to
>     equations, e.g., by moving equations to be inline. We would also make use of
>     most of the ninth page for adding experimental details and results, especially
>     those that the reviewers requested.
> 3. Attached here are graphs that show the training progression of the focus layer
>     vs. transformers and Mega in the associative recall task with a context length of
>     1K, both by epoch https://imgur.com/RPvcAD9 and by clock time
>     https://imgur.com/uvSJLt9. These graphs show a very clear advantage.
>     We note that (i) the tasks we have used are considered extremely challenging.
>     The associative recall experiments depict a clear case in which transformers fail
>     and the recent long-range methods are able to perform better. (ii) we have
>     much fewer resources than researchers at Meta (Mega), Stanford University,
>     Mila, and Université de Montréal (Hyena [Poli et al., 2023]), but are still able to
>     outperform. (iii) in terms of real-world data we show that our method is able to
>     surpass other state-of-the-art language modeling models that have two orders
>     of magnitude more parameters.
>     With regards to peak memory consumption, following the reviewer’s request,
>     see below a comparison over the association-recall task with 1K tokens (maximal fit in
>     memory for transformers)
> | Model | Inference Speed | Memory |
> |----------------------------|-----------------|---------|
> | Transformer | x1 | x1 |
> | S4 [Gu et al., 2022] | x1.58 | x0.43 |
> | MEGA [Xuezhe et al., 2023] | x1.49 | x0.57 |
> | Focus | x1.75 | x0.38 |
> 4. We will make sure to fix all citations to a standard format.

---

### Official Review · Reviewer_K69V · 2023-08-07

**Soundness:** 3

**Excitement:**

4: Strong: This paper deepens the understanding of some phenomenon or lowers the barriers to an existing research direction.

**Paper Topic And Main Contributions:**

This paper proposed a new input dependent layer, which use IIR filters. The experimental results shows it's on par with GPT-2 but with much less parameters on Text8 using BPC metric. The paper also claim it the layer is grounded in the theory of control systems, similar to state-space layers.

**Reasons To Accept:**

The proposed method are new and well motivated. Section 4 give good analysis of stability of IIR.
If proposed method solid, the potential impact are huge.

**Reasons To Reject:**

(1) Experimental results missing some details. I'd suggest adding flops in table 2 and 3. Also adding inference/training cost. For the reported GPT-2 and other paper's results, does it trained on the same data as focused layers?

(2) Easy to scale is one of best property for transformer. Does the new focused layer had similar properties?

(3) Section 5.3 was unclear. The accuracy is 98.8, does it still make sense to compare on this dataset? How should I interpret it? It seems only 0.1% better than previous method.

**Reproducibility:**

4: Could mostly reproduce the results, but there may be some variation because of sample variance or minor variations in their interpretation of the protocol or method.

**Reviewer Confidence:**

3: Pretty sure, but there's a chance I missed something. Although I have a good feel for this area in general, I did not carefully check the paper's details, e.g., the math, experimental design, or novelty.

---

> ### Author Rebuttal · Authors · 2023-08-29
>
> Thank you for the comprehensive review and the constructive feedback.
>
>
> 1. (i) The results of all baselines in Tab. 2 are taken from Mega [Xuezhe et al., 2023]. The training procedure for Focus (our layer) is exactly
>     the same as for Mega and Transformer XL [Dai et al., 2019]. GPT2 is trained with extra data, and performs zero-shot, as detailed in     [Radford et al., 2019], see L611-L612.
>
>     (ii) FLOPs metric for Tab. 2 is not published for other methods, and thus we are not able to compare between the baselines along this
>      metric. In Fig. 3 the complexity of the Focus method is shown, and we also provide a worst-case run time analysis in L462-L488.
>
>     (iii) We will add training/inference times with the full 9-page version. See for example this graph https://imgur.com/RPvcAD9, which
>      depicts training progression for the associative recall task by epoch, and this one https://imgur.com/uvSJLt9, which is by
>      time in seconds and further emphasizes the clear advantage of our layer.
>
>
> 2. Indeed, the scalability of the transformer is one of its best properties. In terms of scalability in sequence length, the Focus layer is
>     similar to that of transformers, due to its attention layer. As for effectively scaling when the model size (and amount of data) increases,
>     this remains an open question that should be answered by empirical experiments with models containing billions of parameters.
>     Unfortunately, we cannot afford such experiments. However, since the Focus layer contains self-attention layers that are known to
>     scale well, there is no reason to believe it will not scale.
>     On the other end of the spectrum, as we show, the focus layer is much less data-hungry than the transformer.
>
>
> 3. The permuted MNIST is considered a hard task, since spatial information is important when dealing with classification from images.
>     This benchmark has become a must-pass test for long-range models. The results show that our method does not fall behind other
>     state-of-the-art long-range methods (and even slightly better as the reviewer noted). In other benchmarks such as the BPC, our method
>     presents remarkable results given the small model size compared to other methods.

---

### Meta-Review · Area_Chair_ejNN · 2023-09-18

**Recommendation:** 4

**Metareview:**

This work proposes a preprocessing layer with IIR filters to efficiently handle the inputs. Reviewers appreciate the novelty and the structure of this work, with comprehensive backgrounds and results. Therefore, the AC recommends acceptance of this paper to EMNLP.

---

### Decision · Program_Chairs · 2023-10-07

**Decision:**

Accept-Main

**Comment:**

This work proposes a preprocessing layer with IIR filters to efficiently handle the inputs. Reviewers appreciate the novelty and the structure of this work, with comprehensive backgrounds and results. Therefore, the AC recommends acceptance of this paper to EMNLP.